# Efficient 3D Object Recognition from Cluttered Point Cloud

**DOI:** 10.3390/s21175850

**Published:** 2021-08-30

**Authors:** Wei Li, Hongtai Cheng, Xiaohua Zhang

**Affiliations:** 1Faculty of Electronic Information and Electrical Engineering, Dalian University of Technology, Dalian 116024, China; davidlee83@mail.dlut.edu.cn (W.L.); xh_zhang@dlut.edu.cn (X.Z.); 2School of Mechanical of Engineering and Automation, Northeastern University, Shenyang 110167, China

**Keywords:** object recognition, point cloud, SAC-IA, RANSAC

## Abstract

Recognizing 3D objects and estimating their postures in a complex scene is a challenging task. Sample Consensus Initial Alignment (SAC-IA) is a commonly used point cloud-based method to achieve such a goal. However, its efficiency is low, and it cannot be applied in real-time applications. This paper analyzes the most time-consuming part of the SAC-IA algorithm: sample generation and evaluation. We propose two improvements to increase efficiency. In the initial aligning stage, instead of sampling the key points, the correspondence pairs between model and scene key points are generated in advance and chosen in each iteration, which reduces the redundant correspondence search operations; a geometric filter is proposed to prevent the invalid samples to the evaluation process, which is the most time-consuming operation because it requires transforming and calculating the distance between two point clouds. The introduction of the geometric filter can significantly increase the sample quality and reduce the required sample numbers. Experiments are performed on our own datasets captured by Kinect v2 Camera and on Bologna 1 dataset. The results show that the proposed method can significantly increase (10–30×) the efficiency of the original SAC-IA method without sacrificing accuracy.

## 1. Introduction

Object recognition and localization are essential functionalities for autonomous robot working in unstructured applications [1]; generally, fast and accurate 3D posture information retrieval is important to tasks, such as material handling, manipulation, assembly, welding, etc [2].

Although 2D image [3]/3D point cloud can all be used to achieve such goal, image-based methods are sensitive to ambient lighting conditions, object texture, and have to estimate 3D information indirectly; hence, point cloud-based methods are becoming promising especially with the emerging of many affordable depth sensors, such as Microsoft Kinect and Intel Realsense cameras.

According to the used features, there are two kinds of pipelines for recognizing 3D objects from point clouds: the global pipeline and the local pipeline. Global feature descriptors, such as ensemble of shape functions (ESF) [4], global fast point feature histogram (GFPFH) [5], Viewport Feature Histogram (VFH) [6], and cluster view feature histogram (CVFH) [7], can be used to evaluate an object observed from different view angles. By generating candidate model clouds all around the object/CAD and extracting their features, the segmented scene cloud can be evaluated and recognized, given some thresholds. The coarse postures can further be refined using Iterative Closest Point (ICP) method [8]. Global feature-based algorithms are efficient in computation time and memory consumption. However, these methods require segmenting the object point cloud from the scene and are difficult to be applied in a cluttered environment with occlusion and complex foreground and background [9].

On the contrary, the local feature descriptors only model the surface characteristics around a single point. Because the points are generally located all around the object and cannot be blocked entirely, hence, they are more robust in occlusion and cluttered environment. The local pipeline mainly consists of keypoint detection, feature matching, hypothesis generation, and evaluation.

Johnson et al. [10] proposed a pipeline using “Spin image (SI)” to identify the objects. SI is a data level shape descriptor which transforms the point cloud into a stack of 2D spin images, and then uses them to represent the 3D model. By matching the model/scene spin images, and filtering and grouping these correspondences, the transformation can be established.

Guo et al. [11] introduced a local descriptor Rotational Projection Statistics (RoPS) to recognize object in the scene, by constructing local reference frames and projecting neighbor points on the reference plane and using statistical values to describe the local feature. It is robust to noise and varying mesh resolution. By selecting some feature points and calculating their feature descriptors, the candidate models are evaluated one-by-one to recognize possible objects.

Rusu proposes Point Feature Histograms (PFH) [12] and Fast Point Feature Histograms (FPFH) [13] descriptors and Sample Consensus Initial Alignment (SAC-IA) initial align method for point cloud registration and object recognition. The use of histograms makes the descriptor invariant view port differences and is robust to noises, hence its widely being used in a lot of registration and matching tasks (usages).

In order to improve the performance, Buch proposes to use both 2D image data and 3D contextual shape data to increase the quality of the correspondences [14]. Results show that the efficiency is significantly improved.

It is noticed that, although local featured-based algorithms have advantage in dealing with occlusion, they require additional time to establish correspondence between model and scene. This process is typically done through Random Sample Consensus (RANSAC). Commonly used initial registration methods for the point cloud mainly include SAC-IA and Super4PCS. Both of these algorithms are implemented based on RANSAC [15]. The RANSAC algorithm mainly uses the idea of multiple sampling iterations to obtain the optimal solution. The algorithm can provide reliable parameter estimates when there are abnormal points in the sample data. The algorithm is highly robust, so it is widely used in machine vision. The field of image registration and object recognition. However, if the number of abnormal points exceeds half, the number of iterations will increase exponentially. These shortcomings will make the algorithm difficult to be applied in actual engineering.

To overcome above shortcomings, some researchers have improved the classic RANSAC algorithm in terms of model solving, sample pre-test, sample selection, and post-stage optimization. The improved method for model solving is mainly for the improvement of the loss function. It includes the following: M-Estimation [16], LMedS [17] MLESAC [18,19], Mapsac [20], etc. M-Estimation is a method that adds weight to data and converts the model solving problem into a least-squares problem with weight added. LMedS uses the method of optimizing which requires the square of the data points residual and minimizing the median value to obtain the optimal solution of the model. However, the above two methods are useful only when the proportion of external points are less than half. Torr et al. used the distribution of inner point/outer point to evaluate the assumption and proposed a method based on maximum likelihood estimation to express the loss function MLESAC to find the optimal solution of the model. After that, the MLESAC method was improved by Torr et al., who proposed Mapsac method. In the improved method, the loss function has been described by Bayesian and maximum posterior probability. The new RANSAC method improves the robustness of the classic algorithm.

For the sample pre-test, the improvement is mainly checking the sample and eliminating the bad sample. The computing time can be decreased by this improvement. RANSAC-Tdd [21] adds the pre-test step and tests the accuracy of the model parameters that are built from the pre-sample. Mates uses conditional probability to describe the quality of each matching point, as well as uses the threshold to select the model, and it proposes the RANSAC-SPRT algorithm [22,23]. The ARRSAC algorithm uses interior points to generate additional hypothesis evaluations after model parameter evaluation. This method uses a competitive evaluation framework to reduce the running time of local optimization, and it improves the efficiency of the algorithm [24].

For the sample selection, it can be divided into sorting method (PROSAC [25]) and positional relationship method (NAPSAC [26] and GROUPSAC [27]), etc., according to the selection strategy.

For the post-stage optimization method, the current optimal solution is used as a starting point to further optimize the model to reduce the number of iterations and improve the efficiency and stability of the algorithm. Some famous methods include LO-RANSAC [28] and OPTIMAL-RANSAC [29].

With the development of machine learning, some researchers combine the reinforcement learning and deep learning with RANSAC. DSAC [30] AND NG-SAC [31] have been proposed to improve the efficiency and accuracy.

In summary, to improve the efficiency and accuracy of point cloud coarse registration, and further improve the three-dimensional object recognition algorithm based on local features, it is necessary to start with the improved RANSAC algorithm.

SAC-IA algorithm is a commonly used architecture for local feature-based methods. However, its efficiency and accuracy is limited in complex cluttered environment. Therefore, the goal of this paper is to improve its performance. The paper analyzes the most time-consuming part of the SAC-IA algorithm: sample generation and evaluation, and proposes two improvements to increase efficiency. The contributions are summarized as follows.

In the initial aligning stage, instead of sampling the key points, the correspondence pairs between model and scene key points are generated in advance and chosen in each iteration, which reduces the redundant correspondence search operations.A geometric filter is proposed to prevent the invalid samples to the evaluation process, which is the most time-consuming operation because it requires transforming and calculating the distance between two point clouds. The introduction of the geometric filter can significantly increase the sample quality and reduce the required sample numbers.

## 2. The Original SAC-IA Approach

A diagram of the SAC-IA is shown in Figure 1. The modifications lie at the sampling and fine refinement process. Instead of sampling the keypoints, the correspondence is sampled. In addition, a geometric constraint is proposed to filter the “impossible” samples in advance to increase the efficiency. In order to overcome the problem brought by a small object in a large scene and avoid to stuck in local minimum, an adaptive ROI extraction operation is proposed to balance restrict the ICP in a local area of the scene point cloud.

### 2.1. Keypoints Detection

Typically, there are thousands of points in a point cloud. Recognizing an object with all of them is complex and not necessary. A commonly used approach is to focus on the more distinctive points instead of ordinary ones. The simplest keypoints detection method is uniform subsampling. However, this operation does not consider the distinctive information among the points; hence, the obtained keypoints are not informative and lack in repeatability.

Several more sophisticated keypoints detection methods are available. Narf 3D keypoint detector [32] is a method of extracting keypoints in the range image. It can only be used in range image captured from a single perspective and cannot be used in point cloud generated from multiple view angles. Harris 3D keypoint detector and SIFT Keypoints [33] follow similar ideas as their 2D versions. Harris 3D uses surface normals to replace the image gradient, while SIFT 3D still has the scaling and rotation invariant features. ISS3D keypoints [34] is an effective keypoints detection method, which mainly uses eigenvalues to select key points. A local coordinate frame is defined for each point using its neighbors within a radius $r_{frame}$. A 3 × 3 scatter matrix is generated, and the eigenvalue $\lambda_{1},\lambda_{2},\lambda_{3}\left(\lambda_{1}>\lambda_{2}>\lambda_{3}\right)$ and vectors are obtained. Then, keypoints can be picked out by evaluating those eigenvalues with some thresholds
(1)$$\begin{array}{c} \lambda_{2}^{2}/\lambda_{1}^{2}<\gamma_{21}\\ \lambda_{3}^{2}/\lambda_{2}^{2}<\gamma_{32} \end{array}.$$

In this paper, ISS3D keypoints is chosen because of its efficiency and robustness (details in comparison can be found in Reference [35]).

### 2.2. Local Shape Descriptors

In order to identify and estimate the pose of an object, it is necessary to find correspondence between model point cloud and scene point cloud. Different from global feature descriptors, local shape descriptors encode features of a point instead of the whole point clouds. Most of the descriptors use the histogram to achieve viewport invariance. They accumulate geometric or topological measurements into histograms according to a specific domain (e.g., point coordinates, geometric attributes) to describe the local surface.

The “spin image (SI)” [11], introduced by Johnson and Hebert, is a typical spatial distribution histogram-based descriptor. The “spin image” descriptor uses the 2D data to represent the 3D feature, which counts the 3D points in a surrounding supporting volume, and then produces a 2D histogram. Therefore, this method will lose valuable 3D shape information.

Signature of Histogram of Orientations (SHOT) [36] encodes histograms of the surface normals in different spatial locations. Therefore, it has the advantage of rotational invariance and robustness to noise. In addition, Prakhya made improvements by converting the real value vector to the binary value vector based on SHOT descriptor and named Binary Signatures of Histogram of Orientations (B-SHOT) [37].

Rusu et al. [13] proposed Point Feature Histogram (PFH) descriptor. This descriptor calculates the normal vector angle and distance between any two points in the search radius of the keypoints to generate the histogram. This descriptor has a computational complexity of $O(nk^{2})$, hence its not being suitable for real-time application. Fast Point Feature Histogram (FPFH) is proposed by Rusu [14] to resolve this problem; FPFH only encodes the relation between the querying point and its k neighborhood points. Therefore, its computational complexity is O(nk). FPFH descriptor is a robust and efficient feature to characterize the local geometry around a point. In this paper, it is used to measure the similarity between different keypoints.

### 2.3. Initial Alignment

For a solid object, at least three corresponding points are required to estimate the transformation matrix between the model and scene point cloud. It is not easy to find the correct match due to similarities in different key points and uncertainties in the point clouds. Although the number of keypoints is fewer than the original point cloud, the search space is still large. Assume there are *M* model key points and *N* scene keypoints, and, for each model keypoint, *K* candidates are considered; the total search space is $M\left(M-1\right)\left(M-2\right)K^{3}\approx M^{3}K^{3}$, which is 125,000, even with 10 keypoints and 5 candidates.

Compared to brute grid search, the random search is more efficient. One such coarse registration method for 3D point clouds is Sample Consensus Initial Alignment (SAC-IA) [14]. It consists of the following four steps:Keypoints Sampling. Select three sample points from the model point cloud, and ensure that their distances are greater than the user-defined minimum distance $d_{min}$.Correspondence Searching. For each sample point, find a list of points (K candidates) in the scene point cloud with a similar local descriptor. This is usually done by searching KD-Tree. Finally, randomly choose one point from the candidates as the correspondence pair.Transformation Matrix Estimation. With three correspondence pairs, it is able to estimate the transformation matrix $T_{j}$, where *j* is the sampling iteration index.Performance Evaluating. Transfer the model point cloud with $T_{j}$ and compare it against the scene point cloud. Compute an error metric based on those two point clouds using a Huber penalty measure
(2)$$E_{j}=\sum_{i=1}^{n}H(l_{i})1,$$
where $E_{j}$ is the overall measure,
(3)$$H\left(l_{i}\right)=\left\{\begin{array}{cc} \frac{1}{2}l_{i}^{2} & ∥l_{i}∥<m_{i}\\ \frac{1}{2}m_{i}\left(2∥l_{i}∥-m_{i}\right) & ∥l_{i}∥>m_{i} \end{array},\right.$$
and $m_{i}$ is a predefined threshold used to exclude the outliers, while $l_{i}$ is the minimum distance between a model point and scene point cloud.

Finally, by repeating the above four processes, the true correspondence is assumed to definitely occur. The robustness is proportional to the trial number. However, too many trials will decrease the efficiency; hence, there is a trade-off between stability and efficiency. In spite of that, SAC-IA can still get a good result after many iterations and be widely used in the initial alignment process.

### 2.4. Fine Alignment

SAC-IA randomly chooses three corresponding points in a model and scene point cloud and obtains the best guess for the object location. However, due to the uncertainty in keypoint extraction and sensors, the obtained transformation matrix may deviate from the real one. Refinement has to be performed to the aligned model and scene with all points involved.

Iterative Closest Point (ICP) [9] is the commonly used registration method. It is based on point-to-point registration and uses LM to calculate the transformation matrix. ICP has good precision when aligning two similar point clouds with a small initial posture error. If the initial posture differs too much, it may stuck in local minimum. Therefore, typically, ICP follows a curse alignment method, such as SAC-IA.

### 2.5. The Overall Workflow of Standard SAC-IA Algorithm

The above process can be summarized in the following Algorithm 1. By providing model point cloud and scene point cloud, the object pose is returned in the form of a transformation matrix.
**Algorithm 1:** Standard SAC-IA Pose Estimation Algorithm.**Input:** 
 The model point cloud: $PC_{m}$The scene point cloud: $PC_{s}$**Output:** 
The transformation matrix of the best match: $T^{*}$1:Downsampling $PC_{m}\to {\hat{PC}}_{m}$ and $PC_{s}\to {\hat{PC}}_{s}$2:Detecting keypoints $KP_{m},KP_{s}$3:Calculating FPFH descriptors $PD_{m},PD_{s}$4:**for**j=1 to $n_{max}$ **do**5:    Randomly choose three points $P1_{m}^{j},P2_{m}^{j},P3_{m}^{j}$6:    Filter samples by distance threshold $d_{min}$7:    For $Pi_{m}^{j}$, find $n_{K}$ similar points $Pi_{s}^{j}$ from $KP_{s}$8:    Randomly choose one $Pi_{s}^{j}$ for $Pi_{m}^{j}$9:    Estimate the transformation $T^{j}$ with $Pi_{m}^{j}\to Pi_{s}^{j}$10:    Transform ${\hat{PC}}_{m}$, count inliers with threshold $d_{max}$11:**end for**12:Find the transformation matrix ${\bar{T}}^{*}$ with most inliers13:Using ICP to find the optimal transformation $T^{*}$14:**return** 
$T^{*}$

It is noted that the model processing, keypoint detection, and PFPH calculation can be done offline. Therefore, the scene point processing and the loops 3–8 take most of the computation time.

### 2.6. The Problems

The standard SAC-IA algorithm depends on RANSAC to estimate the model. Some limitations of conventional RANSAC will affect the efficiency.

Random sampling:The sampling is performed on the keypoint level. For a typical configuration, a model set includes 50,000 points. To find their correspondence points, 5 candidates are explored. This means KD-tree search operation is performed totally, which will result in 750,000 times.Transform matrix estimation and evaluation:In standard SAC-IA, the models are represented as a transformation matrix. One has to optimize the matrix parameters and then transform all point clouds and calculate their distances to the scene point cloud. Therefore, this estimation and evaluation process is time-consuming. Moreover, this operation is repeated for $n_{max}$ times for robustness consideration, which greatly restricts its efficiency

## 3. The Efficient SAC-IA Approach

### 3.1. The Workflow of Improved SAC-IA Algorithm

The efficiency is limited by the sampling process. In order to overcome this difficulties, two main aspects are improved. (1) Instead of picking samples from the key points, the correspondence pairs are randomly selected. (2) A geometric constraint is proposed to filter the “impossible” samples in advance, to increase the efficiency. The Efficient approch is improved on the basis of Algorithm 1, see below Algorithm 2.
**Algorithm 2:** Efficient SAC-IA Pose Estimation Algorithm.**Input:** 
 The model point cloud: $PC_{m}$The scene point cloud: $PC_{s}$**Output:** 
The transformation matrix of the best match: $T^{*}$1:Downsampling $PC_{m}\to {\hat{PC}}_{m}$ and $PC_{s}\to {\hat{PC}}_{s}$2:Detecting keypoints $KP_{m},KP_{s}$3:Calculating FPFH descriptors $PD_{m},PD_{s}$4:For each keypoint in $KP_{m}$, Build correspondence map $C_{ij}=<P_{m_{i}},P_{s_{i}j}>,i=1,\cdots ,n_{m},j=1,\cdots ,n_{K}$5:**for**k=1 to $n_{max}$ **do**6:    **repeat**7:        Choose $C_{k}=\left[C_{i_{1}j_{1}},C_{i_{2}j_{2}},C_{i_{3}j_{3}}\right]$;8:        Filter $C_{k}$ by threshold and triangle similarity9:    **until** $C_{k}$ is valid10:   Estimate the transformation $T^{k}$ with $C_{k}$11:**end for**12:Find the transformation matrix ${\bar{T}}^{*}$ with most inliers13:Using ICP to find the optimal transformation $T^{*}$14:**return** 
$T^{*}$

### 3.2. The Sampling Process

In standard SAC-IA process, the sampling is performed on the keypoint level. Keypoints are randomly chosen firstly and then try to find their correspondences based on FPFH similarity. Usually, three points are used to estimate the transformation matrix. In order to increase the robustness, usually, $n_{k}$ candidates are explored. This means the KD-tree search operation is performed for a total of $n_{max}\times n_{k}\times 3$ times.

It is noticed that much of these queries are redundant. As shown in Figure 2, since there are totally $n_{m}$ keypoints in the model point cloud, the possible correspondence pairs are $n_{m}\times n_{K}$. Usually, $n_{m}$ is significantly smaller than $n_{max}$; hence, the above cost can be eliminated by changing the sampling target and sequences. If there are 100 keypoints in the model point cloud and $n_{k}=5$, the search queries only 500 times.

Based on this observation, the following sampling method is proposed.

One Shot Correspondence Map Building. Before the random sampling iteration loop, for each keypoint $P_{m_{i}}$ in $KP_{m}$, search their $n_{K}$ most similar keypoints $P_{s_{ij}}$ in $KP_{s}$ based on FPFH descriptor. This is done by performing nearest neighbor search in the KD-tree. Then, a correspondence map $C_{ij}=<P_{m_{i}},P_{s_{ij}}>,i=1,\cdots ,n_{m},j=1,\cdots ,n_{K}$ can be derived.Correspondence Level Sampling. Instead of sampling keypoints, the correspondence pairs are sampled directly. Since there is no need to evaluate their similarities, the sampling efficiency can be accelerated. In each iteration, three pairs $C_{k}=\left[C_{i_{1}j_{1}},C_{i_{2}j_{2}},C_{i_{3}j_{3}}\right]$ are randomly picked out. These samples need to be filtered before evaluation. First of all, their indices should not be equal to each other; second, their distances should be bigger than a threshold $d_{min}$. Third, those three points should not be in a line. In addition to the above constraints, this paper proposes a new filtering criteria to improve the correspondence quality: Triangle Similarity Constraint-based sample filter.

### 3.3. Triangle Similarity Constraint Based Sample Filter

In order to reduce the matching time, the samples have to be selected carefully before sending to estimation and evaluation. Standard SAC-IA only considers the distances between model keypoints by setting a minimum threshold. This can ensure that the sample points are far apart, which is beneficial for the matrix estimation process. However, this condition is too loose to filter out “bad” samples. These points can fulfill the filter conditions, but these points and their similarities in the scene cannot calculate the transformation matrix.

In this paper, another constraint is proposed based on triangle similarity. Because, in each iteration, three correspondences are selected which involve three model points and three scene points, they are expected to correspond to each other in pairs. As shown in Figure 3, this assumption leads to a fact that there are two triangles in the model and scene, and they should be similar/identical, i.e., their side lengths should be the same.

This geometric constraint is based on solid basis and can filter out many invalid samples. A direct benefit of filtering out many “bad” samples is avoiding many useless transformation matrix estimations and evaluations, increasing the quality of the samples and decreasing the iteration number $n_{max}$.

The Triangle Similarity-based sample generation and filtering process is as follows:Sampling: Three pairs $C_{k}=\left[C_{i_{1}j_{1}},C_{i_{2}j_{2}},C_{i_{3}j_{3}}\right]$ are randomly picked out.Removing Same Points: $i_{1}\neq i_{2},i_{1}\neq i_{3},i_{2}\neq i_{3}$, $j_{1}\neq j_{2},j_{1}\neq j_{3},j_{2}\neq j_{3}$.Filtering Close Samples: $\left|P_{m}^{i_{1}},P_{m}^{i_{2}}\right|>d_{min}$, $\left|P_{m}^{i_{1}},P_{m}^{i_{3}}\right|>d_{min}$, $\left|P_{m}^{i_{2}},P_{m}^{i_{3}}\right|>d_{min}$.Filtering Dissimilar Samples: In order to increase the stability, the following measure is proposed to evaluate the similarity:
(4)$$\begin{array}{ccc} d_{ij} & = & \left|∥P_{m}^{i_{1}},P_{m}^{i_{2}}∥-∥P_{s}^{j_{1}},P_{s}^{j_{2}}∥\right|\\ \; & + & \left|∥P_{m}^{i_{1}},P_{m}^{i_{3}}∥-∥P_{s}^{j_{1}},P_{s}^{j_{3}}∥\right|\\ \; & + & \left|∥P_{m}^{i_{2}},P_{m}^{i_{3}}∥-∥P_{s}^{j_{2}},P_{s}^{j_{3}}∥\right| \end{array}.$$If $d_{ij}>d_{similar}$, this sample will be removed, and a resampling process is performed until a valid sample is acquired.

$d_{min}$ and $d_{similar}$ are two parameters used in the candidate pair filtering process. They are tuned according to the sensor performance and object size. $d_{min}$ should not smaller than the object radius and large enough to ensure the pairs can make robust estimation of the object posture. $d_{similar}$ should be small enough to filter out more “bad” samples. However, if it is too small, because of sensor noises, it may also reduce the possibility of finding potential objects. Therefore, in real application, these two parameters have to be tuned accordingly.

## 4. Experiments

### 4.1. Experimental Setup

A Kinect v2 Camera is used to capture the point clouds of object. The algorithm is implemented using C++ and Point Cloud Library (PCL) in Windows 10. The computer is equipped with an Intel Core i7-8650U CPU (2.11 GHz, 4 cores) and 16 GB memory.

A stackable storage bin is used as the test object, whose CAD is shown in Figure 4. By placing the box randomly on the table and grabbing it with different poses (with occlusion), point clouds are captured, as shown in Figure 4. Totally, there are 17 scene point clouds in the dataset.

### 4.2. Efficiency Verification Experiments

The proposed Efficient SAC-IA (E-SAC-IA) and original SAC-IA were both implemented for comparison. The used parameters are listed in Table 1.

The CAD file is converted into a point cloud by subdividing the triangular meshes; therefore, the point density is not even. To resolve this issue, a fine resolution is firstly used and then downsampled with a leaf size of 0.01 m, to match the scene point clouds captured with real sensor.

In the implementation of Efficient SAC-IA, instead of using specific keypoint detector, the generic subsampling operation is used. By setting the leaf size as 0.04 m, a sparse point cloud is generated from the original density one. The remaining points are treated as keypoints. For model point cloud, there are 160 key points, while, for the scene point clouds, typically, 370 keypoints are available.

#### 4.2.1. Performance against Max Iteration Number

Max iteration number is the key to control the robustness and efficiency of the RANSAC-based algorithms. For the proposed E-SAC-IA, the performance changes against max iteration number are evaluated. The 17 scene point clouds are fed into the algorithm with different max iteration numbers ( 50; 100; 200; 300; 400; 500; 600). The results are given in Figure 5.

Figure 5a shows the success rate and recognition time when changing the max valid sample number from 50 to 600. It is seen that, with the increase of max valid sample number, the success rate increases. When 500 more samples are used, the success rate is above 90% percent. In addition, the recognition time is proportional to the sample number. The average recognition time is around 200–300 ms.

Figure 5b shows the ratio between total samples and valid samples. The valid samples are samples that fulfill the minimum distance and triangular similarity constraints. It is found that the ratio is 100:1 and nearly keeps constant with the change of max valid sample number parameters.

Due to the randomness and variations between each scene point cloud, the actual recognition time may changes. Figure 5c shows that mean and standard division of the recognition time. The standard division is 30 ms.

In comparison, the original SAC-IA is also tested. The max sample number is chosen as 50,000, to guarantee the success rate. The success rate is 0.941, while the recognition time is shown in Figure 5d. It can be found that the recognition time is around 10 s, which is nearly 30 times that of the proposed E-SAC-IA (Algorithm 2).

#### 4.2.2. Performance against Sampling Order

In spite of the iteration number in the RANSAC procedures, the sampling order and type also affect the efficiency. As previously mentioned, in conventional SAC-IA algorithm, the key points are randomly selected in each iteration, and then, by finding their nearest neighbors through Kd-Tree search, the correspondence is established online. This nearest neighbor search is queried for the same times as the iteration number, which is significantly larger than the keypoints number. Therefore, by moving the correspondence building process outside the loop and randomly selecting point pairs, this issue can be eliminated.

In Figure 5d, the recognition time of three configurations is given. The original SAC-IA with sample number is 50,000, and the E-SAC-IA with sample number is 500 (equivalent to about 50,000 total samples). The last one is the original SAC-IA with the above improved sample selection process, but without sample filters.

From the results, one can see that reorganizing the sampling order can also improve the overall efficiency. Compared to original SAC-IA, it saves about 2.5 s for 50,000 iterations, i.e., 50 μs for each nearest neighbor search operation.

### 4.3. Test on Bologna 1 Dataset

In order to verify the performance on objects with different shapes, the algorithm is tested on Bologna 1 dataset [37]. These datasets, created from some of the models belonging to the Stanford 3D Scanning Repository, are composed of 6 models (armadillo, buddha, bunny, Chinese dragon, dragon, and statuette, as shown in Figure 6) and 45 scenes. Each scene contains a subset of the 6 models that are randomly rotated and translated. The 45 scenes belong to three types: 3 objects, 4 objects, and 5 objects. Each type has 15 scenes.

In this experiment, ISS algorithm is used to detect the keypoints. The configuration is listed in Table 2.

For each scene, no matter whether it contains specific model, it is processes to detect all those models. The detection time is averaged and listed in Table 3. Because of the ISS keypoint detector, the keypoints are less than previous experiments. Therefore, the overall runtime is lower. With the increase of points in the scene, the detection time increases slightly. The recognition time is below 200 ms for the proposed E-SAC-IA method, while, for the original SAC-IA method, the recognition time is around 2000 ms.

The acceleration and success ratio are listed in Table 4. One can find that it is within the range of 11–18 times.

One can find that the success rate is identical for original and improved SAC-IA algorithms. In most cases, the rate is above 90%. As shown in previous experiments, the success rate is related to several issues, such as the max iteration number, the quality of the keypoints, and fuzziness of the scene point cloud.

Therefore, it can be concluded that the proposed E-SAC-IA method indeed can increase the object recognition efficiency without scarifying the success rate. For typical configuration, the recognition time is about 200–300 ms, i.e., 3–4 fps. Therefore, it can be applied in some real-time applications. Because the method still follows the local pipeline, the keypoint detection, feature calculation, and ICP restriction all cost some time (about 150 ms), and the space to decrease the recognition time is bounded if no GPU acceleration or multiple core paralleling techniques are used. The acceleration ratio is related to the keypoints number, randomness, and point numbers. According to the experimental results, the typical ratio is 10× to 30×.

## 5. Conclusions and Future Works

This paper studies the problem of recognizing 3D objects from cluttered scenes with point cloud information. The bottleneck to the efficiency of the conventional SAC-IA method is identified: the sampling target and the sampling criteria. Accordingly, two improvements are made to reduce the sampling time. Firstly, the correspondences between model keypoints and scene keypoints are established beyond the sampling loop to reduce the feature matching and nearest neighbor search operation. Considering the magnitude difference between correspondence number and max sampling iteration number, this change can save amount of computing time; secondly, a triangle similarity-based geometric constraint is proposed to filter out “bad” samples to prevent them for further evaluation. Considering the computational complexity of sample evaluation operation, this filtering mechanism can significantly reduce the sampling time and increase the sampling efficiency.

Although the efficiency is in some degree improved, the robustness of the method is still restricted by the keypoint detector, feature descriptor and sample filter. In the future, RGB/texture information will be incorporated to increase the distinction between keypoints in the model and environment; The intensity/color/depth keypoints are fused and filtered to provide more robust keypoints for correspondence estimation.

## Figures and Tables

**Figure 1 sensors-21-05850-f001:**
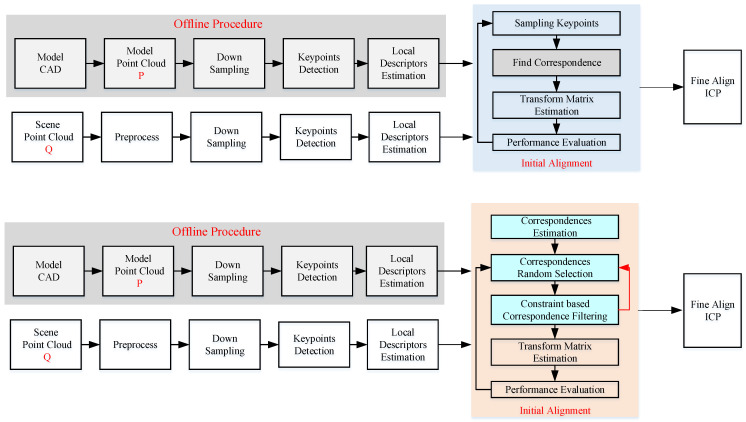
Diagram of the original SAC-IA algorithm and the proposed efficient SAC-IA algorithm.

**Figure 2 sensors-21-05850-f002:**
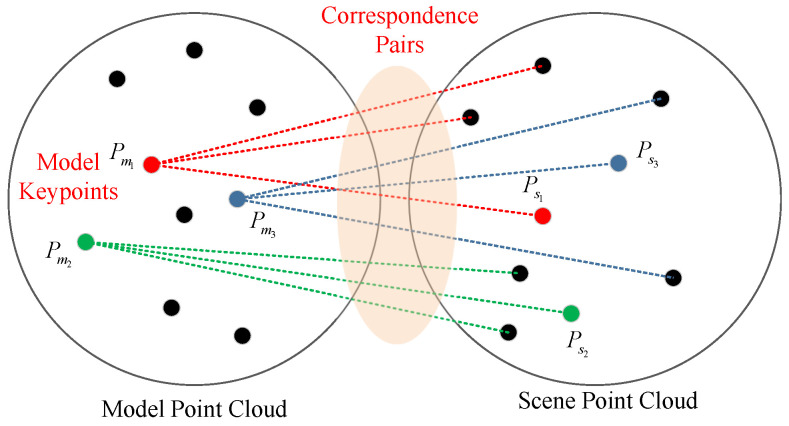
Finding the correspondence pairs in scene point cloud.

**Figure 3 sensors-21-05850-f003:**
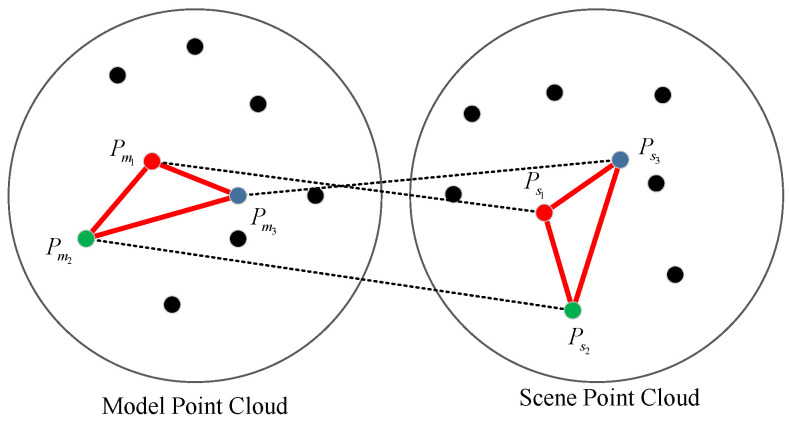
The assumption of geometric constraint, two similar triangles in a model and scene point cloud.

**Figure 4 sensors-21-05850-f004:**
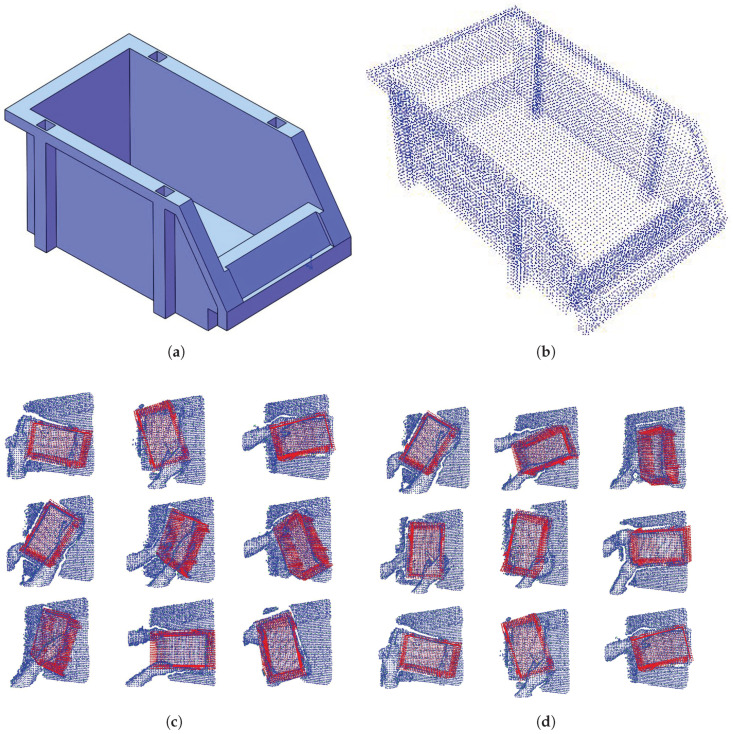
The object CAD model and recorded point cloud datasets. (**a**) The CAD model. (**b**) Point cloud model of the object; different pose with occlusion is shown in (**c**,**d**).

**Figure 5 sensors-21-05850-f005:**
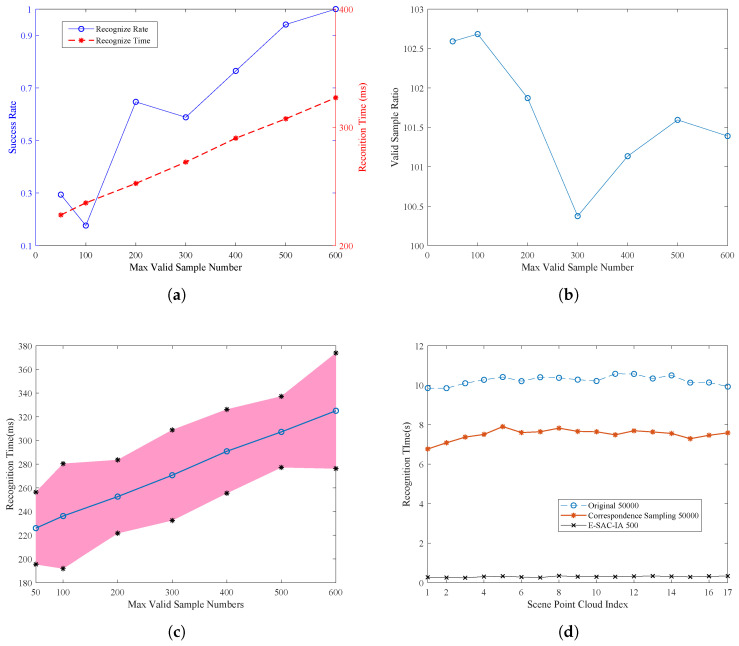
The perfomace comparison between the traditional and improved SAC-IA algorithm.

**Figure 6 sensors-21-05850-f006:**
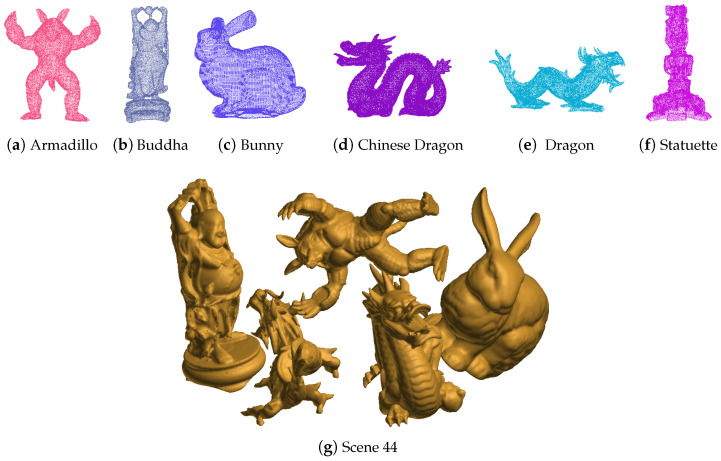
The object models in the Bologna 1 datasets.

**Table 1 sensors-21-05850-t001:** The used parameter values in the Algorithm.

Parameter	Value
Voxel Leaf Size (m)	0.01
Downsample Size (m)	0.04
Number of Points in Model (subsampled)	2101
Number of Points in Scene (subsampled)	≈6000
Number of Model Keypoints	160
Number of Scene Keypoints	≈370
Randomness (K)	5
Triangle Similarity Threshold $d_{similar}$ (m)	0.06
Minimum Sample Point Distance $d_{min}$ (m)	0.03
Max Iteration Number (proposed)	50; 100; 200; 300; 400; 500; 600
Max Iteration Number (original)	50,000
Thread number for OpenMP	4

**Table 2 sensors-21-05850-t002:** The used parameters in the Bologna experiments.

Parameter	Value
Voxel Leaf Size (m)	0.005
ISS Salient Radius (m)	0.018
ISS Non-Max Radius (m)	0.012
Number of Points in Model (subsampled)	2014; 2635; 3017; 3929; 1913; 2279
Number of Points in Scene (subsampled)	≈8000; ≈11,000; ≈14,000
Number of Model Keypoints	30; 46; 42; 44; 17; 48
Number of Scene Keypoints	≈110; ≈140; ≈180
Randomness (K)	5
Triangle Similarity Threshold $d_{similar}$ (m)	0.1
Minimum Sample Point Distance $d_{min}$ (m)	0.03
Max Iteration Number (proposed)	1000
Max Iteration Number (original)	10,000
Thread number for OpenMP	4

**Table 3 sensors-21-05850-t003:** Recognition time of E-SAC-IA and Original SAC-IA Algorithm.

Model Name	E-SAC-IA Runtime (ms)			Original SAC-IA Runtime (ms)		
	Scene 3	Scene 4	Scene 5	Scene 3	Scene 4	Scene 5
armadillo	116.8	127.9	133.3	1905.6	1950.0	1979.0
buddha	170.0	180.0	183.4	2423.4	2471.0	2487.5
bunny	158.2	169.7	174.7	2290.6	2319.6	2359.9
Chinese Dragon	169.5	169.6	178.4	2333.3	2377.2	2428.6
Dragon	76.0	84.8	88.5	1500.0	1576.0	1587.3
statuette	183.0	193.0	197.4	2447.0	2508.1	2557.5

**Table 4 sensors-21-05850-t004:** Efficiency and success ratio of the E-SAC-IA to original SAC-IA Algorithm.

Model Name	Scene 3	Scene 4	Scene 5	Success Rate of SAC-IA(%)	Success Rate of E-SAC-IA(%)
armadillo	15.31	14.24	13.84	100	100
buddha	13.25	12.72	12.56	96.42	100
bunny	13.47	12.66	12.50	100	100
Chinese Dragon	12.76	13.01	12.61	100	100
Dragon	18.73	17.58	16.93	82.14	82.14
statuette	12.37	11.99	11.95	91.30	91.30

## Data Availability

Not applicable.

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
