# Peer review of "Efficient 3D Object Recognition from Cluttered Point Cloud"

_sensors, 2021, doi:10.3390/s21175850_

Round 1
Reviewer 1 Report
Some keywords should be added.
The first sentence should start with a capital letter.
The quotation marks used are from different charsets. Unicode character 8220 is used for the left quotation mark and ASCII character 32 is used for the right quotation character instead of Unicode character 8221.
The reference in the text indicated by the question mark ([?] at line 78) should probably have been [21]. Reference 15 is not used at all.
The initial alignment in Fig. 1 looks like an endless loop, some decision block that would end the initial alignment process should be added.
The notation of the conditions determining the thresholds should either be reformatted to be in the text line or should be numbered as an equation.
Some typographic errors should be checked e.g. spaces before and after parentheses (136, Table1), missing periods (135)…
Chapter 2.6 is too vague, problems should be clearly defined and explained.
Some claims are not sufficiently substantiated by references or described experiments. E.g. 263, 213÷215
The object in Fig. 4 and line 291 is a stackable storage bin, not a container box. Individual scenes at the figure should be rearranged and/or resized to better fit the page. Also, markups (a, b, c…) mentioned in the figure description are missing in the figure itself.
There are problems with the spelling of some words e.g. derivate/deviate (197), propotional/proportional (316), standard devision/division (323, 324)…
Oxford comma is missing at lines 351, 368, 376, 392.
The font size variates between tables. I think that tables 4 and 5 could be merged.
There are many problems with the use of singular and plural in the article.
Reviewer 2 Report
The reviewed manuscript, entitled “Efficient 3D Object Recognition from Cluttered Point Cloud” is dedicated to presenting the standard SAC-IA algorithm and the author-proposed improvements to improve its performance: pre-generating the correspondence pairs between model and scene key points and proposing a geometric filter to prevent the invalid samples to the evaluation process. It presents an intriguing approach, supported by the standard, yet dedicated experiments, and it leads to interesting remarks. The experiments are described well, the advantage of the proposed approach is highlighted reasonably. The paper is written and organized well, the overall structure of the manuscript is proper.
Some remarks are pointed out:
1. The manuscript requires modifications, introduced to the use of English grammar and text syntax: e.g. sentences start with non-capital letters, spaces are not provided between reference number and text.
2. Error is present in the Author’s list. The e-mail address is displayed between the Authors’ names, and the same affiliation is displayed twice
3. References are sometimes improper. MAPSAC is missing reference in line 78, the paper by Torr et al. is missing reference in line 83, some more issues are also present.
4. Some minor issues with figures are present, the quality of several figures is not satisfactory.
5. In Figure 1, the offline procedure is the same in both approaches. Can it be not shown, or shown only once? Maybe instead of presenting the postulated approach over the initial one, only the initial should be displayed, with the changed elements crossed out and replaced by new elements, proposed by the Authors? However, implementing this remark is not mandatory.
6. The description of Figure 1 should be significantly reduced. Only the first sentence should be used. The rest should be given in the normal text.
7. In figure 4, the (a) to (e) markers are not present in the picture, only in the figure title.
8. The descriptions in Figure 6 are not necessary. They should be transferred to the figure title or excluded.
To sum up, in the Reviewer’s opinion, the publication of the paper should be recommended, after addressing the abovementioned minor issues. The provided results should be interesting for the research community, and the topic of the paper is in the scope of the “Sensors” journal.
